# Exploring the Complexity of the Human Respiratory Virome through an In Silico Analysis of Shotgun Metagenomic Data Retrieved from Public Repositories

**DOI:** 10.3390/v16060953

**Published:** 2024-06-13

**Authors:** Talya Conradie, Jose A. Caparros-Martin, Siobhon Egan, Anthony Kicic, Sulev Koks, Stephen M. Stick, Patricia Agudelo-Romero

**Affiliations:** 1Wal-Yan Respiratory Research Centre, Telethon Kids Institute, Perth, WA 6009, Australia; 2Medical, Molecular and Forensic Sciences, Murdoch University, Perth, WA 6150, Australia; 3Centre for Computational and Systems Medicine, Health Future Institute, Murdoch University, Perth, WA 6150, Australia; 4Department of Respiratory and Sleep Medicine, Perth Children’s Hospital for Children, Perth, WA 6009, Australia; 5Centre for Cell Therapy and Regenerative Medicine, School of Medicine and Pharmacology, Perth, WA 6009, Australia; 6School of Population Health, Curtin University, Perth, WA 6102, Australia; 7Perron Institute for Neurological and Translational Science, Perth, WA 6009, Australia; 8Centre for Molecular Medicine and Innovative Therapeutics, Murdoch University, Perth, WA 6150, Australia; 9Australian Research Council Centre of Excellence in Plant Energy Biology, School of Molecular Sciences, The University of Western Australia, Perth, WA 6009, Australia; 10European Virus Bioinformatics Centre, Friedrich-Schiller-Universitat Jena, 07737 Jena, Germany

**Keywords:** respiratory viruses, shotgun metagenomics, lung virome, genome assembly, viromics, microbiome, airway, omics

## Abstract

Background: Respiratory viruses significantly impact global morbidity and mortality, causing more disease in humans than any other infectious agent. Beyond pathogens, various viruses and bacteria colonize the respiratory tract without causing disease, potentially influencing respiratory diseases’ pathogenesis. Nevertheless, our understanding of respiratory microbiota is limited by technical constraints, predominantly focusing on bacteria and neglecting crucial populations like viruses. Despite recent efforts to improve our understanding of viral diversity in the human body, our knowledge of viral diversity associated with the human respiratory tract remains limited. Methods: Following a comprehensive search in bibliographic and sequencing data repositories using keyword terms, we retrieved shotgun metagenomic data from public repositories (n = 85). After manual curation, sequencing data files from 43 studies were analyzed using EVEREST (pipEline for Viral assEmbly and chaRactEriSaTion). Complete and high-quality contigs were further assessed for genomic and taxonomic characterization. Results: Viral contigs were obtained from 194 out of the 868 FASTQ files processed through EVEREST. Of the 1842 contigs that were quality assessed, 8% (n = 146) were classified as complete/high-quality genomes. Most of the identified viral contigs were taxonomically classified as bacteriophages, with taxonomic resolution ranging from the superkingdom level down to the species level. Captured contigs were spread across 25 putative families and varied between RNA and DNA viruses, including previously uncharacterized viral genomes. Of note, airway samples also contained virus(es) characteristic of the human gastrointestinal tract, which have not been previously described as part of the lung virome. Additionally, by performing a meta-analysis of the integrated datasets, ecological trends within viral populations linked to human disease states and their biogeographical distribution along the respiratory tract were observed. Conclusion: By leveraging publicly available repositories of shotgun metagenomic data, the present study provides new insights into viral genomes associated with specimens from the human respiratory tract across different disease spectra. Further studies are required to validate our findings and evaluate the potential impact of these viral communities on respiratory tract physiology.

## 1. Introduction

The human body hosts a diverse array of colonizing microorganisms, including bacteria, fungi, archaea, protozoa, and viruses, collectively forming ecological communities referred to as the human microbiota [1]. The composition of the human microbiota varies between different organ systems, with inter-individual differences arising from various factors, such as environmental exposure and genetic influences [1]. The observed variability in the microbiota’s functional and taxonomic composition has also been linked to some diseases causally associated with various pathogenic processes in humans [2]. However, due to the simplicity of profiling bacteria using universal marker genes, much of the research on the human microbiota has predominantly focused on its bacterial component, with other components, such as viruses, remaining largely understudied [3].

The term “virome” refers to a subpopulation of the microbiota, which includes different types of viruses that can be classified depending on their host. This includes viruses that infect eukaryotic cells, and bacteriophages, which are capable of infecting bacteria [4,5,6]. Both eukaryotic viruses and bacteriophages can impact human health in both harmful and beneficial ways, including triggering immune responses or influencing residential bacterial communities within the same compartments [4,6]. In the last few decades, emerging infectious diseases caused by viruses, such as severe acute respiratory syndrome coronavirus (SARS-CoV-1), H1N1 influenza virus (Swine flu), Middle Eastern respiratory syndrome coronavirus (MERS-CoV), and severe acute respiratory syndrome coronavirus 2 (SARS-CoV-2), have become increasingly recognized as major threats to public health. Therefore, cataloguing existing viromes and understanding host–virus interactions are of essential value to protecting both individual and population level health [4,6,7,8].

While the lungs were initially considered a sterile organ, emerging evidence indicates the presence of a distinct microbiota [9]. The lower airways host a transient microbial population that likely originates in the oronasal cavity and reaches the lungs via micro-aspiration [10]. As a result, it is probable that respiratory epithelia are continually exposed to various clinically relevant viruses [9,11]. Viruses from multiple families, including bacteriophages, have been identified within the respiratory tract [4]. Certain viral families have shown associations with specific clinical groups; for instance, bacteriophages are frequently found in samples from individuals with cystic fibrosis, and *Anelloviridae* are prevalent in specimens from lung transplant recipients [5,7]. Eukaryotic viruses impact human infections and their progression, while bacteriophages can shape resident bacterial populations and alter functions of bacterial genomes by facilitating DNA transfer between cells, contributing to the spread of antimicrobial resistance [4,12,13].

Current lung virome studies are often subjected to unstandardized methodologies, leading to difficulties in viral identification. Up to 95% of sequences in different studies remain uncharacterized due to methodological variations [4,5,7,14,15,16]. This uncharacterized portion is referred to as “viral dark matter” and can be attributed to the incapability of databases that were generated using limited shotgun metagenomic data and often fail to assign sequences to known taxonomy. With metagenomic sequencing technologies rapidly improving, large datasets are being generated. However, computational strategies and current reference databases have not expanded at the same pace, limiting the ability to characterize viral sequences [17,18]. It is widely accepted that our current understanding of viral classification and taxonomy greatly underestimates the true diversity of viruses. The main public repository for viral sequences is the National Centre for Biotechnology (NCBI) database, but it tends to be biased as it does not encompass the full diversity of the viral community. Additionally, available reference sequences are biased towards those that can be cultivated and are clinically relevant, narrowing the scope of identifiable viruses. This leads to purely novel and unidentified viruses remaining uncharacterized, and those slightly different from reference genomes potentially being mischaracterized [17,18]. Before exploring new samples and datasets, revisiting existing databases and thoroughly characterizing the available sample sets can expand diversity and reinforce current databases. This strategy has been employed in various human body sites, such as the skin and the gut, to address the issue of “viral dark matter” [19,20,21]. With lung virome research lagging in terms of available material, accessing and reanalyzing historical data are critical for a better understanding of true lung virome diversity. The aim of this study was to collate publicly available sequence data from human respiratory samples and reanalyze them with the objective of providing a more comprehensive understanding of viral diversity associated with the respiratory tract.

## 2. Materials and Methods

### 2.1. Bibliographic Search and Data Collection

Two strategies were used to collate lung virome studies for which sequencing data was publicly available (Figure 1). Strategy 1 involved a literature search on PubMed, utilizing specific keywords related to “lungs”, “sputum” or “lavage fluid”, “next generation sequencing” or “metagenomics”, “bioinformatics”, and “lung virome” (Appendix A), and identified 18 studies with associated sequencing data obtained from the indicated repositories (Figure 2). Strategy 2 involved exploring sequence data repositories from NCBI Sequence Read Archive (SRA), NCBI Gene Expression Omnibus (GEO), and EMBL-EBI European Nucleotide Archive (ENA), using specific keywords related to “lung virome”, “bacteriophages”, “metagenomics”, and “shotgun sequencing” to capture bioprojects that may contain unique information not yet linked to a published study. This second strategy yielded 67 unique studies (Figure 2).

### 2.2. Curation of Study Information

A total of 85 bioprojects were identified and linked to their original published studies, which were used to extract methodological information and metadata. Key study details, including title, authors, citation, year, and methodological specifics, were entered into our in-house database. Subsequently, a methodological screening of the database was conducted (Appendix A).

Lung or respiratory biofluid samples were categorized as saliva, nasopharyngeal swabs and aspirates, sputum, bronchoalveolar lavage (BAL) fluid, and lung tissue. Studies were excluded under the following conditions: those using samples of non-human origin, biological material unrelated to the lung/respiratory tract, studies focused solely on the bacterial microbiome or fungal mycobiome using targeted amplicon or enrichment library preparation methods that would exclude viral sequences, studies focused on characterizing specific viral genomes, or those lacking identification in the original manuscript. Only those studies linked to their original papers or those providing sufficient methodological information in bioproject-associated metadata were retained. Ultimately, 43 studies met the screening criteria and were further analyzed to establish a curated database of the airway virome (Figure 2).

Each of the bioprojects associated with these 43 studies was classified based on study design, sequencing technology, study purpose (diseases or healthy clinical phenotypes), subject age, sample type, and sample size (Appendix A). Additionally, the following were considered: sequencing platforms (e.g., short-read or long-read), types of reads (single- and/or paired-end), sequencing strategy (DNA, RNA, or both), and the nucleic acid extraction kits and library preparation techniques used. These methodological features were sourced from the materials and methods sections and the supplementary data within each study. In cases where discrepancies were identified between the information in the peer-reviewed published article and the sequence repository, the data available in the sequence repository were prioritized.

### 2.3. Retrieving Raw FASTQ Files

FASTQ files were downloaded from specific repositories using the software fastq-dl (v1.0.6, accessed on 1 April 2022, https://github.com/rpetit3/fastq-dl). A total of 43 bioprojects containing 868 independent sequencing files were downloaded.

Issues were identified with six of the studies from the finalized database, which included data not being available for download, data not correctly linked to the respective bioproject, or the absence of FASTQ files for download despite their stated availability [22,23,24,25,26,27]. To address this, formal requests were made via email to both the SRA, and the first and corresponding authors of the respective bioprojects. As a result, two bioprojects were successfully incorporated into the database. In one case, the SRA released the data for a bioproject [22], and in another, a linkage error was corrected [23]. Additionally, although two bioprojects were not accessible, the authors of those projects provided direct access to their data [24,25].

### 2.4. Bioinformatic Analysis

Data were processed using the bioinformatic pipeline EVEREST (https://github.com/agudeloromero/EVEREST, v0.01, accessed on 1 April 2022). EVEREST is an end-to-end pipeline designed for virus discovery, structured into five main phases that use FASTQ files as input. Briefly, during the pre-processing phase, files undergo quality control through trimming [26,27], followed by a filtering phase that includes host removal, replicated sequences elimination, and digital normalization [28,29,30]. Next, a de novo assembly is constructed using SPAdes [31] and similar contigs are clustered [32]. In the refinement phase, viral contigs are captured with VirSorter2 [33] and their quality is assessed using CheckV [34]. Finally, during the viral classification phase, two databases are used, nucleotide (NCBI) and amino acid (Uniprot), to taxonomically classify the viral contigs [32,35]. Each process is executed by select and specific software tools organized within the pipeline itself, as illustrated in Appendix A.

The data output produced by EVEREST was used for taxonomic classification. Each individual FASTQ file within a bioproject generated its own output files. Therefore, data was isolated and compiled separately for each sequencing file and then aggregated for each bioproject. Subsequently, these bioproject-level datasets were consolidated into a single metadata file.

The compiled data included both qualitative and quantitative information. Qualitative data included taxonomic and Baltimore classification, classification as a provirus, and quality levels as per CheckV [34]. Quantitative information included metrics such as GC content, contig length, and RPKM (Read Per Kilobase of reference sequence per Million total sequencing reads).

### 2.5. Statistical Analysis

Statistical analysis was performed in R v4.0.2. The normality of data distribution was evaluated using a Shapiro–Wilk test, and a Wilcoxon rank sum test was performed to compare groups when normality assumptions were not met. When conducting multiple comparisons, type I errors were controlled for by adjusting *p*-values using the Bonferroni correction method. Principal component analysis models were generated using a mixOmics R package [36]. Ecological alpha diversity estimates and PERMANOVA analysis were performed using the functions of R package Vegan (https://github.com/vegandevs/vegan, v2.6-4, accessed on 30 May 2024). The cut-off for statistical significance was set at *p*-value < 0.05.

## 3. Results

### 3.1. Study and Data Demographics

Following rigorous screening of the current literature and public repositories, a total of 85 lung virome metagenome studies were compiled, of which 43 were selected after manual curation (Appendix A). Out of the 43 bioprojects, 30 contained paired-end sequencing data and were processed through our bioinformatic pipeline, EVEREST, with 16 providing results (Table 1 and Appendix A). The remaining 13 projects utilized a single-end sequencing strategy and were not processed because, at the time of the study, EVEREST was not equipped to handle this sequencing format. Among the 30 reanalyzed bioprojects, there were a total of 868 FASTQ sequencing files. EVEREST successfully detected viral contigs in 194 (22%) of these FASTQ files, which were associated with 16 different bioprojects (refer to Data Availability Statement, Table 1 and Appendix A).

The 194 FASTQ files with successful viral contigs represented 523 subjects, including 131 adults, with 84 of them presenting a respiratory pathology, and 392 children, with 384 of them diagnosed with acute respiratory infection (Table 1 and Appendix A). In some studies, individual FASTQ files represented pooled samples from different individuals, which explains the discrepancy between the sequencing files and number of subjects represented in those files. The 523 individuals represent 13 different groups based on clinical phenotypes provided in the manuscript, including different respiratory pathologies, such as acute respiratory infection, cystic fibrosis, sarcoidosis, and lung transplant recipients (Table 1 and Appendix A). The studies represented a range of biological specimens from the upper and lower respiratory tract, with bronchoalveolar lavage (BAL) fluid, sputum, throat swabs, and nasopharyngeal aspirates proving to be the most successful sample types for viral contig detection (Table 1 and Appendix A).

### 3.2. Viral Contig Classification

EVEREST identified a total of 1842 contigs based on quality, of which 1414 were assigned taxonomy. Among these, 146 (8%) were classified as proviruses and excluded from further analysis, while the remaining 1696 (92%) were classified as viruses (Appendix A). EVEREST annotated 1193 (84%) contigs as bacteriophages, and the remaining 221 (16%) were identified as eukaryotic viral contigs (Figure 3A). Contigs were also grouped following the Baltimore classification system [52]. Most of the bacteriophage contigs (610, 51%) were classified as Group I, corresponding to double-stranded DNA bacteriophages. Six bacteriophage contigs (0.5%) were classified as Group II, representing single-stranded DNA bacteriophages. The remaining bacteriophage contigs could not be classified within the Baltimore system as they were not classified taxonomically to the family level. Eukaryotic viral contigs were distributed across four specific Baltimore groups, with the highest proportion (83, 37%) identified as Group I, followed by Group II (43, 20%), Group IV (20, 9%), and Group V (7, 3%) (Figure 3B).

Bacteriophage taxonomy was annotated to the order level (513, 43%), family level (366, 30%), subfamily rank (13, 1%), genus level (64, 5%), and species level (71, 6%). The remaining bacteriophage contigs were assigned to higher taxonomic ranks, including superkingdom (64, 5%) and clade (15, 1%), with 87 contigs (7%) unassigned to taxonomy (Figure 4A). Eukaryotic viral contigs were classified at the superkingdom (47, 21%), family (51, 23%), genus (36, 16%), or species (46, 21%) level. The remaining contigs were annotated at different taxonomic ranks or unassigned (15, 7%) (Figure 4A).

Viral contigs annotated at least down to the family level were distributed across 25 different families, including one unclassified family. Bacteriophages were distributed across 11 families, with the majority classified into the families *Siphoviridae* (359, 30%), *Myoviridae* (103, 9%), or *Podoviridae* (24, 2%). EVEREST also annotated 27 contigs as CrAssphages or CrAss-like phages (Appendix A), a type of bacteriophage commonly observed in human fecal metagenomes [53]. These contigs were associated with six FASTQ files from three different bioprojects (PRJNA392272, PRJNA419524, and PRJNA494633) (Appendix A) [14,41,43]. Eukaryotic viral contigs were distributed across 19 families, including *Phycodnaviridae* (18, 8%), *Siphoviridae* (17, 8%), *Herpesviridae* (16, 7%), *Microviridae* (13, 6%), *Papillomaviridae* (13, 6%), *Picornaviridae* (12, 5%), *Anelloviridae* (10, 4%), and *Papillomaviridae* (10, 4%) (Figure 4B).

### 3.3. Assessment of Viral and Proviral Genomes

EVEREST implements CheckV [34] to evaluate the quality of the assembled contigs and ranked them based on their completeness in comparison to assessed reference genomes. The 1842 identified contigs were categorized into five quality tiers [34], with the majority being classified as low quality (0–50% completeness; 1111, 60%) or undetermined quality (for which a completeness estimate was available; 448, 24%). The remaining contigs were identified as medium-quality (50–90% completeness; 137, 7%), high-quality (>90% completeness 83, 4%), and complete (63, 3%) genomes (Figure 5A).

Among the 146 contigs classified as complete or high-quality genomes, most were annotated up to order (47, 32%) and family (34, 23%) levels, with 21 (14%) annotated as either unclassified or no rank (Figure 4B). Complete and high-quality contigs were assigned to 15 different viral families, with most of these contigs annotated as unclassified families within the Caudovirales order (47, 32%), followed by *Microviridae* (20, 14%), while 22 (15%) contigs were unclassified (Figure 5C).

To confirm that there was no bias in contig assembly associated with the different bioprojects, GC content and length were evaluated for the assembled genome for the 1414 eukaryotic and bacteriophage viral contigs assigned to taxonomy down to the family level (Figure 6). Contigs were distributed in a bimodal distribution regarding their GC content, with values ranging from 0.2 to 0.7. The contig length distribution was right-skewed, with values ranging from 5000 to 130,000 bp (Figure 6). Contigs annotated to specific families were clustered around similar GC content values, while contig length was neither associated with specific viral families nor linked to altered GC values, suggesting no bias in contig generation across different bioprojects.

For GC content, compared to families with a GC content under 0.4, some families were found to cluster in uniform sections, showing a uniform GC content of these viral families even between samples (Figure 6). In terms of genome length, some families showed strong and partial clustering, indicating that these families showed consistency in the length of their genomes between various studies and cohorts (Figure 6).

### 3.4. Metanalysis of Viral Communities in the Respiratory Tract

We next explored the viral community profiles obtained through EVEREST in relation to covariates, such as disease state, respiratory niche, or the type of biospecimen used to sample the airways. We excluded from these analyses the profiles obtained from the two reports studying 18th-century mummified lung tissue from subjects infected with *Mycobacterium tuberculosis* as the representativeness of viral particles and nucleic acids in these samples may be compromised due to their age of over 200 years [38,50]. Principal component analysis (PCA) was used to explore viral profiles in the samples included in this study for which viral contigs were obtained (Figure 7). The first two components of the PCA model explained 30% of the variance associated with our dataset, with an unclassified family within Caudovirales and *Siphoviridae* being the major drivers of variation along the first two components (Figure 7A).

In our dataset, samples obtained from adults were associated with both an unclassified family within Caudovirales and *Siphoviridae* (Figure 7A,B). Interestingly, samples obtained from children clustered around the center of the PCA, indicating that they were not associated with the major source of variation represented by components 1 and 2 (Figure 7A,B). Likewise, samples from healthy individuals were associated with *Siphoviridae*, while those with a respiratory pathology showed overall positive correlation with an unclassified family within Caudovirales and negative correlation with *Siphoviridae* (Figure 7A,C). On the other hand, samples from the lower airways were mostly associated with an unclassified family within Caudovirales, while samples from the upper airways were mainly associated with the family *Siphoviridae* (Figure 7A,D). Variations in the virome of lung transplant recipients were associated with members of an unclassified family within Caudovirales (Figure 7A,E). Likewise, the abundance of members from an unclassified family within Caudovirales was associated with the virome of BAL samples (Figure 7A,F).

To understand the impact of the defined variables, including age, disease type, and biospecimen type, on the composition of viral communities represented in the PCA model, a permutational multivariate analysis of variance (PERMANOVA) analysis was performed. Our findings revealed that the origin of the samples, denoted by the bioproject from which the samples were obtained, had a substantial effect on the overall viral community structure represented in the PCA model (PERMANOVA, *pseudo* F = 3.90, R^2^ = 0.23, *p*-value = 0.0001). Likewise, based on the sum of squares (10.9% explained by the type of sample, and 22.8% explained by the disease group), the results of the PERMANOVA analysis suggest that, within our dataset, both biospecimen type (PERMANOVA, *pseudo* F = 3.51, R^2^ = 0.11, *p* = 0.0002) and disease groups (PERMANOVA, *pseudo* F = 4.50, R^2^ = 0.23, *p*-value = 0.0001) had a strong effect over the observed structure of the viral communities associated with the respiratory tract. Interestingly, when consolidating the different disease groups into a single “disease” category, the impact of the binary variable, disease or healthy, on the overall structure of the respiratory virome in our dataset was less pronounced compared to the analysis that considered the individual disease states (PERMANOVA, *pseudo* F = 2.93, R^2^ = 0.02, *p*-value = 0.003), perhaps suggesting disease-specific viral community profiles. Similarly, other predefined groups, such as patient age categories (adults or children) and respiratory tract niche (upper or lower) associated with each of the respiratory samples incorporated into the PCA model, had a small effect on the overall structure of the resulting viral communities (PERMANOVA for age, *pseudo* F = 4.01, R^2^ = 0.02, *p*-value = 0.0001; PERMANOVA for respiratory niche, *pseudo* F = 5.71, R^2^ = 0.03, *p* = 0.0001).

The alpha diversity estimate, as measured by the Shannon index, was assessed and compared across different sample-associated descriptors, including biospecimen origin, age group category, and clinical phenotype, using pairwise comparisons (Figure 8). For this analysis, we excluded viral profiles from bioprojects PRJNA419524 and PRJEB32062 [14,37] as they represent longitudinal samples and were, therefore, analyzed separately. Biospecimen origin, whether from the upper or lower respiratory tract, showed no effect on viral diversity (Wilcoxon rank sum test (WRST), *p*-value = 0.09) (Figure 8A). Similarly, age group category was not associated with viral diversity (WRST, *p*-value = 0.06) (Figure 8B). Conversely, samples collected from healthy individuals demonstrated higher diversity than those subjects diagnosed with a respiratory pathology (WRST, *p*-value = 0.0007) (Figure 8C). To gain further insights into this association, viral diversity in the context of specific disease conditions was explored. Compared to healthy controls, acute respiratory infections (WRST with Bonferroni correction, *p*-value = 0.004) and lung transplant recipients (WRST with Bonferroni correction, *p*-value = 0.002) were associated with lower viral diversity in the respiratory tract (Figure 8D). On the contrary, patients with sarcoidosis demonstrated comparable diversity values to that of healthy controls (Figure 8D). Lastly, the impact of the type of sampling on viral diversity was examined (Figure 8E). Compared to bronchoalveolar lavage (BAL) fluid, viral diversity in nasopharyngeal aspirates (WRST with Bonferroni correction, *p*-value = 0.04), sputum (WRST with Bonferroni correction, *p*-value = 0.02), and throat swab samples (WRST with Bonferroni correction, *p*-value = 0.002) were higher (Figure 8E). In contrast, viral diversity in nasopharyngeal swabs was comparable to that of BAL fluid (Figure 8E).

Bioproject PRJEB32062 contains sequencing files representing longitudinal sputum samples from patients with cystic fibrosis, which were obtained during both exacerbation episodes and periods of clinical stability [37]. After running EVEREST, we obtained longitudinal viral profiles from three patients. As shown in Appendix A, viral diversity in these samples was highly variable over time, with exacerbation episodes associated with both high and low viral diversity values even in samples obtained from the same patient (Appendix A). Bioproject PRJNA419524 contains sequencing files from BAL fluid obtained from donor lungs and recipients, the latter being monitored longitudinally [14]. EVEREST reported viral profiles from 40 FASTQ files within this dataset representing 13 LTR subjects (Appendix A). Longitudinal data were available for 11 LTR subjects, of which 7 LTR subjects also had viral profiles from the donor lungs (Appendix A). As shown in Appendix A, viral alpha diversity was largely equivalent between BAL collected from donor lungs and the first BAL collected post-transplant from the recipient (mean difference 0.39, 95% confidence interval [−0.54, 1.32]) (Appendix A). Likewise, we did not observe differences in BAL-associated viral diversity between the first two consecutive timepoints collected post-transplant (mean difference 0.15, 95% confidence interval [−0.42, 0.72]) (Appendix A).

## 4. Discussion

In this study, we reanalyzed sequence data files from biological specimens obtained from the human respiratory tract to better understand viral composition and diversity associated with the human airways. Although similar studies have been performed in samples from the human gut and skin niches [19,20,21], our study represents a new insight into understanding viral diversity associated with the respiratory system in humans. Our bioinformatic pipeline recovered 1842 viral contigs, with 146 representing either complete or near-complete viral genomes. Taxonomy was assigned to 1414 contigs (77%) representing 25 viral families. This classification included both known and unidentified viral families within the human respiratory tract. The analysis of identified contigs unveiled a wide spectrum of viral families, including some complete and high-quality genomes. In general, EVEREST reproduced the viral profiles reported in the original studies, except for one study in which EVEREST did not capture contigs classified within the *Anelloviridae* family. However, while the original study accessed the NCBI database in 2013 [40], EVEREST uses a more updated version [35], which might explain these discrepancies. Interestingly, most of the recovered contigs were classified as bacteriophages (Appendix A). This observation is in line with the observed high diversity of bacteriophages in other human niches such as the gut, in which phages might play an important role in the modulation of the intestinal ecosystem [54]. A more plausible explanation for the high observed diversity of bacteriophages in the reanalyzed studies/datasets is the presence of a specific bacterial host. Accordingly, we observed *Mycobacterium* phages in FASTQ files from both bioprojects containing data from individuals positive for *Mycobacterium* infection (PRJEB7454 and PRJNA189842) [38,50]. Likewise, in bioprojects containing sequencing data from people with cystic fibrosis such as PRJEB32062, phages were detected from pathogens typically found in the airways of patients with cystic fibrosis, such as *Pseudomonas*, *Staphylococcus*, or *Ralstonia* (Appendix A) [37]. Notably, a viral family not previously identified within the lung virome was discovered. Thus, we recovered multiple genomes representing CrAssphages, a group of bacteriophages typically associated with the human gut [19,53,54]. This was an unexpected finding, given that previous lung virome studies have not identified CrAssphages in their datasets [16,26,40,55]. However, the presence of CrAssphage contigs in six different FASTQ files from three different studies [14,41,43] involving samples from both adults and children across various clinical groups strongly suggests that they do not represent contaminating sequences or methodological artifacts specific to individual studies. Although the biological significance of CrAssphages in the airways is not within the scope of our study, these viruses may enter the airways along with gastrointestinal bacteria through reflux-micro-aspiration processes, which are commonly associated with different respiratory pathologies [56]. Overall, our observations suggest that, as it has been recently observed in the gut [19,54], bacteriophage populations constitute a significant component of human respiratory microbiota.

Ecological descriptors were used to assess viral community profiles in relation to covariates, such as disease states, different respiratory niches, or the type of biospecimen used to survey the airways. Most viral community profile variation was explained by the different bioprojects from which the sequencing data were obtained. This finding is not surprising as methodological differences in obtaining viral nucleic acids, such as the introduction of viral enrichment steps, or other aspects, such as sequencing library preparations or sequencing methods, are expected to substantially influence the portion of the virome that will be captured in each study [57]. Similarly, the type of specimen used for sampling the airways and the disease group from which the biological specimen was obtained significantly impacted the resulting viral community profiles. The influence of the type of specimen on viral community composition may be attributed to niche-specific bacterial communities along the respiratory tract [58] or the impact of a higher proportion of host DNA on sequencing sensitivity for virome profiling (e.g., when using lung tissue samples) [59]. Thus, nasopharyngeal aspirates, sputum, and throat swab samples demonstrated higher viral diversity compared to BAL fluid, which showed similar diversity to nasopharyngeal swab samples. On the other hand, variation in viral community profiles associated with different clinical groups may be indicative of disease-specific airway bacterial compositions. For example, viruses belonging to the family *Siphoviridae* were correlated with the virome of healthy subjects, while an unclassified family within the Caudovirales order was associated with lung transplant recipients. Interestingly, while individuals with disease demonstrated reduced viral diversity in the airways compared to healthy controls, this trend was not consistently applicable across the different pathologies evaluated in our study. In contrast to samples obtained from acute respiratory infections or lung transplant recipients, which displayed lower viral diversity in comparison to samples from healthy controls, alpha viral diversity estimates in specimens from patients with sarcoidosis were equivalent to those of healthy individuals. Furthermore, compared to the viral profiles observed in adult samples, the viral communities in samples obtained from children were not associated with specific viral families, even though 98% of these samples were obtained from children with acute respiratory infections. This observation suggests that, as with the bacterial component of the microbiota, there could be either an age-related ecological succession in the viral communities of the human respiratory tract [58,60] or the airway virome in children is more resilient to perturbation.

Our study has several limitations. Firstly, the small number of studies and datasets available at the time of analysis, coupled with the highly heterogenous population related to both the clinical presentation and respiratory niche sampled, represent challenges. Thus, it is important to exercise caution when interpreting the findings of our study as the limited number of available studies and differences in airway sampling methodologies across the various cohorts could have potentially confounded our observations. Furthermore, most of the analyzed studies targeted DNA, with only one study targeting both RNA and DNA populations. Consequently, our study is biased towards DNA viruses, likely underestimating the true diversity of the virome associated with the human respiratory tract.

## 5. Conclusions

In conclusion, our exploratory analysis provides new insights into the viral families present in the human respiratory tract and offers preliminary observations regarding covariates that may influence viral community composition. By leveraging publicly available repositories of sequencing data, the present study provides a more nuanced picture of the viral population associated with the human airways and illustrates how the implementation of a recently developed computational pipeline can be used to characterize viral sequences from shotgun metagenomic sequencing data repositories.

## Figures and Tables

**Figure 1 viruses-16-00953-f001:**
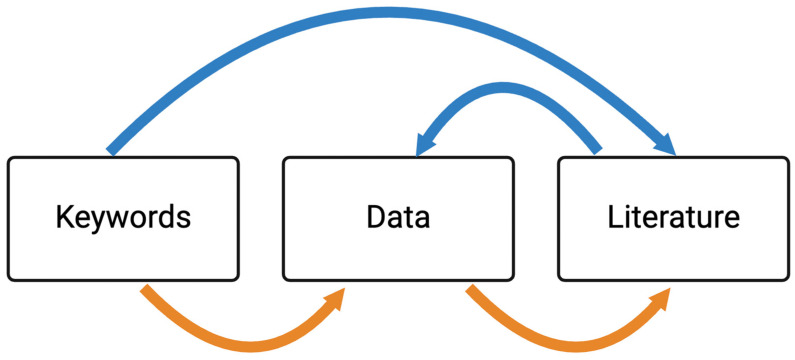
The bibliographic search and data collection strategy used. A flow diagram depicting the two search strategies used for data collection, with the blue arrows representing search strategy 1, and orange arrows representing search strategy 2.

**Figure 2 viruses-16-00953-f002:**
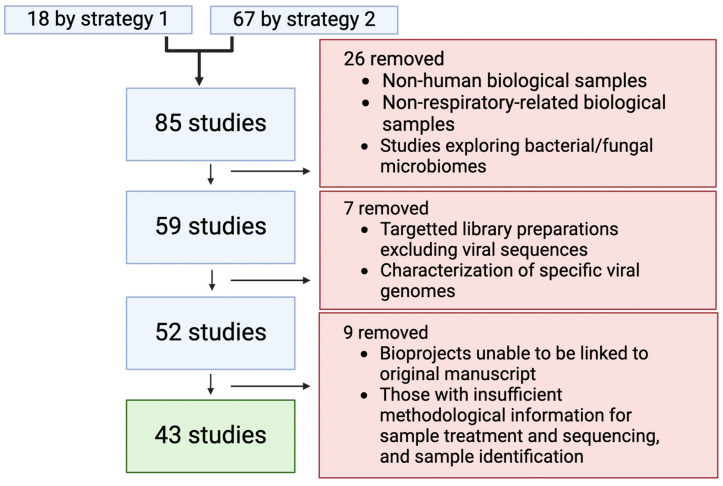
The study-filtering strategy used. The flow diagram graphically represents the search strategy employed to identify relevant sequence data from public repositories. The number of studies excluded at each filtering step and the reasons for excluding these bioprojects are also indicated. A final 43 studies were identified that matched the selection criteria and were used for analysis in the present study.

**Figure 3 viruses-16-00953-f003:**
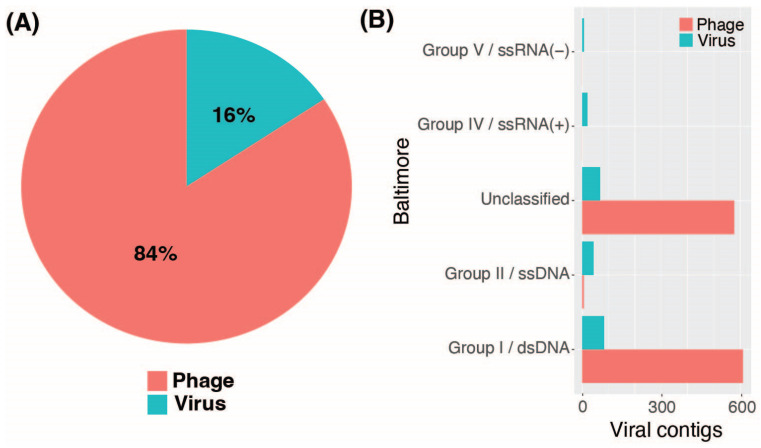
The distribution of eukaryotic viral contigs and bacteriophage contigs captured through EVEREST. (**A**) A pie chart depicting the 1414 captured contigs assigned to taxonomy, from which 1193 (84%) were identified as bacteriophages (red), and 221 (16%) were identified as eukaryotic viruses (blue). (**B**) Baltimore classification of viral contigs. The bar plot summarizes the number of eukaryote virus (blue) and bacteriophage (red) contigs within the indicated Baltimore classes, including those that could not be classified (unclassified).

**Figure 4 viruses-16-00953-f004:**
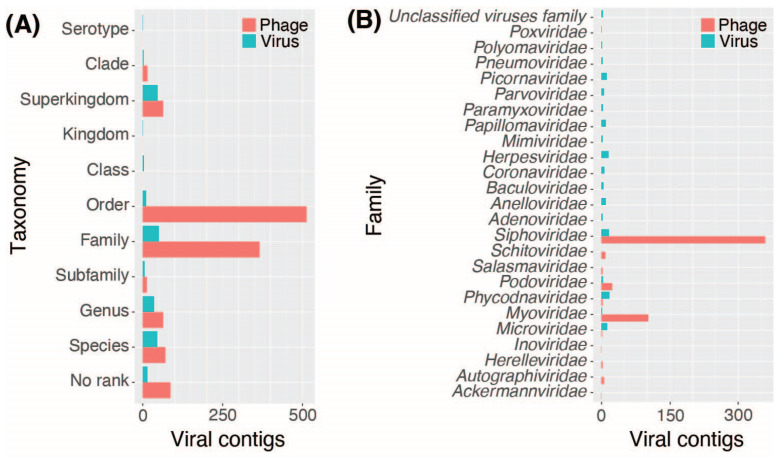
The distribution of taxonomic classification following the lowest common ancestor method included in EVEREST. (**A**) A bar plot showing the number of eukaryotic (blue) and bacteriophage (red) viral contigs classified at the indicated taxonomy ranks. (**B**). Taxonomic classification at the family level. The bar plots represent the 25 families identified among the bacteriophage (red) and eukaryotic (blue) viral contigs.

**Figure 5 viruses-16-00953-f005:**
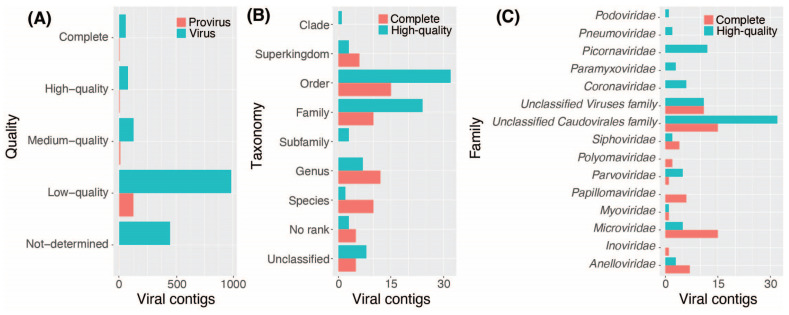
Genome quality assessment of the viral contigs and taxonomic characterization of the complete and high-quality genomes. (**A**) Bar plots represent the distribution of the 1842 contigs classified based on their quality as determined by CheckV [34]. Blue bars represent viral genomes, while red bars represent proviruses. (**B**) Bar plots representing the number of complete (red) and high-quality (blue) genomes that were assigned at each taxonomic rank. (**C**) Bar plots representing the number of complete (red) or high-quality (blue) viral genomes that were assigned to the indicated viral families.

**Figure 6 viruses-16-00953-f006:**
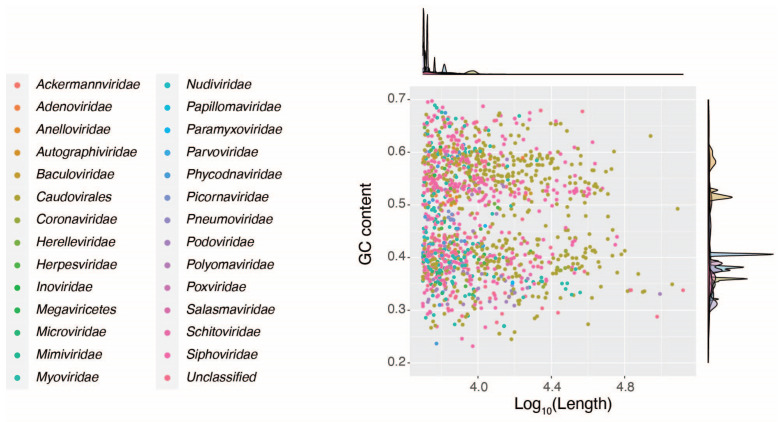
An overview of the viral contigs recovered from the bioprojects analyzed in this study grouped by taxonomic viral families. The scatter plot shows the correlation between GC content and viral contig size. Density plots of viral families on the top and right sides represent contig size and GC content, respectively.

**Figure 7 viruses-16-00953-f007:**
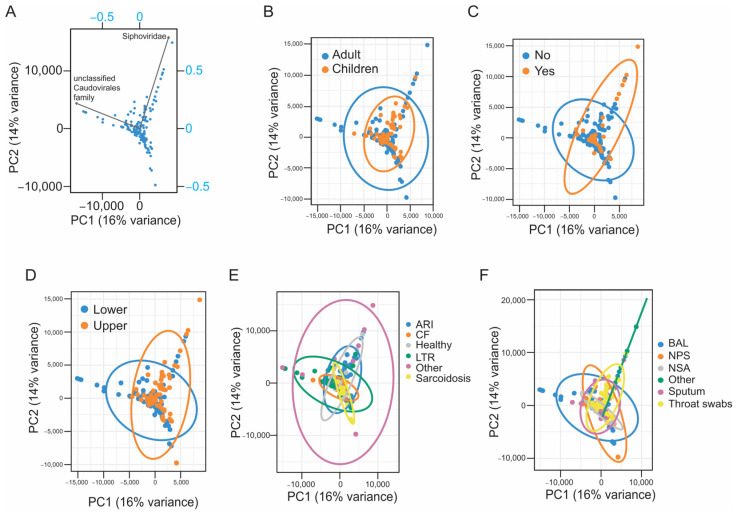
(**A**–**F**) A projection of the samples included in this study in the two first components of the PCA model. (**A**) A biplot showing the PCA score plot and the loading plot. The left and bottom axes represent the PCA scores of the samples, and the top and right axes indicate how strong each feature (vector) influences the principal components. Vectors represent features with at least a 0.6 correlation coefficient with any of the two first components of the PCA model, and their project values on each principal component indicates how much weight they have on those specific components of the PCA model. The unclassified family within Caudovirales is negatively correlated with component 1 and represents the major source of variation along this component. Conversely, *Siphoviridae* exhibit a positive correlation with component 1. Distribution across component 2 is mostly driven by *Siphoviridae* (positive correlation) and to a lesser extent by the unclassified family within Caudovirales. For each PCA graph, dots represent individual samples. Samples are labelled based on whether they were obtained from adults (blue) or children (orange) (**B**), whether they were obtained from healthy individuals (orange) or individuals with disease (blue) (**C**), whether they were obtained from the lower (blue) or upper (orange) respiratory tract (**D**), the specific respiratory pathology (**E**), or the type of specimen used to sample the airways (**F**). ARI: acute respiratory infection; CF: cystic fibrosis; LTR: lung transplant recipient; BAL: bronchoalveolar lavage; NPS: nasopharyngeal swabs; NSA: nasopharyngeal aspirates.

**Figure 8 viruses-16-00953-f008:**
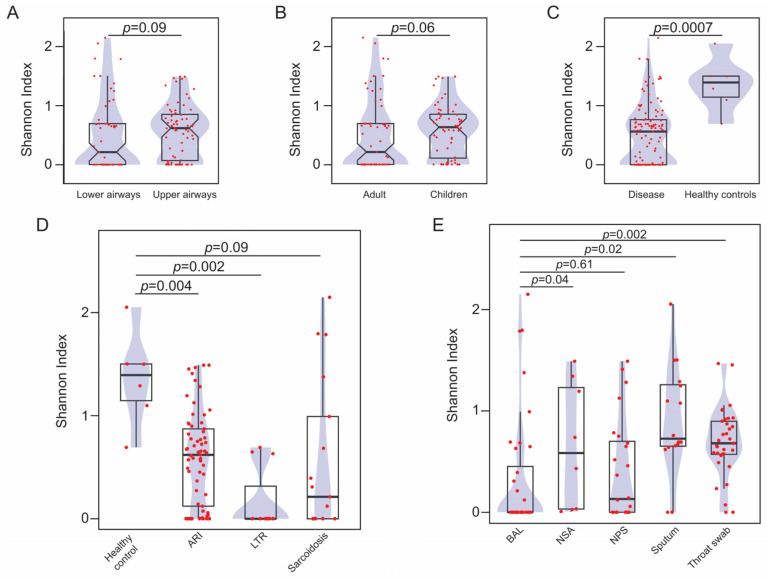
Shannon index diversity of viral families identified in lung samples. (**A**–**E**) Box plots displaying the comparison in alpha viral diversity between samples from the upper and lower respiratory tract (**A**), between samples collected from adults and children (**B**), between samples obtained from healthy controls or individuals with disease (**C**), between samples obtained from patients with different respiratory pathologies and healthy controls (**D**), and between bronchoalveolar lavage fluid and other type of specimens from the respiratory tract (**E**). Red dots represent individual datapoints. Only groups containing more than three datapoints were included in these analyses. The density plots (shaded blue area) represent the probability distribution for each data group. Differences between the indicated groups were evaluated using a Wilcoxon rank sum test and the resulting *p*-values are indicated in each panel. In A and B, notches in the boxplot represent the 95% confidence interval for the median. In D and E, *p*-values were corrected using the Bonferroni method. ARI: acute respiratory infection; LTR: lung transplant recipient; BAL: bronchoalveolar lavage; NSA: nasopharyngeal aspirates; NPS: nasopharyngeal swabs.

**Table 1 viruses-16-00953-t001:** General description of the bioprojects from which EVEREST successfully recovered viral contigs. A more comprehensive data table is provided in Appendix A.

Bioproject	Clinical Group	Library Preparation	Sample Location	Total FASTQ Files	Successful FASTQ Files
PRJEB32062 [37]	CF	DNA	Sputum	25	18
PRJEB7454 [38]	TB	DNA	Mummified lung tissue	9	6
PRJNA316588 [39]	CF, COPD, Healthy smokers	DNA	Sputum	18	11
PRJNA369654 [40]	LTR, HIV	RNA and DNA	BAL	22	14
PRJNA392272 [41]	Sarcoidosis	RNA and DNA	BAL	98	25
PRJNA419524 [14]	LTR	RNA and DNA	BAL	63	40
PRJNA493096 [42]	Respiratory failure	RNA	BAL	4	2
PRJNA494633 [43]	ARI	RNA and DNA	NPS, NPA, Sputum	39	32
PRJNA573045 [44]	URTI	RNA	NPS	4	4
PRJNA601736 [45]	COVID-19	RNA	BAL	2	2
PRJNA623895 [46]	COVID-19	RNA	NPS	1	1
PRJNA629087 [47]	LTR	RNA	BAL	21	1
PRJNA671740 [48]	Lung adenocarcinoma	RNA and DNA	LT	5	1
PRJNA779483 [49]	ARI	RNA and DNA	Throat swabs	37	33
PRJNA189842 [50]	TB	DNA	Mummified lung tissue	1	1
PRJNA639353 [51]	Healthy	RNA	NTS	91	3

ARI, acute respiratory infection; CF, cystic fibrosis; COPD, chronic obstructive pulmonary disease; HIV, human immunodeficiency virus; LTR, lung transplant recipient; TB, tuberculosis; URTI, upper respiratory tract infection; BAL, bronchoalveolar lavage; LT, lung tissue; NPS, nasopharyngeal swabs; NPA, nasopharyngeal aspirates; NTS, nasal-throat swabs.

## Data Availability

The following bioprojects were successfully inputted through EVEREST: PRJEB32062 (https://www.ncbi.nlm.nih.gov/bioproject/PRJEB32062/) (accessed on 1 April 2022), PRJEB7454 (https://www.ncbi.nlm.nih.gov/bioproject/PRJEB7454/) (accessed on 1 April 2022), PRJNA316588 (https://www.ncbi.nlm.nih.gov/bioproject/PRJNA316588/) (accessed on 1 April 2022), PRJNA369654 (https://www.ncbi.nlm.nih.gov/bioproject/PRJNA369654/) (accessed on 1 April 2022), PRJNA392272 (https://www.ncbi.nlm.nih.gov/bioproject/PRJNA392272/) (accessed on 1 April 2022), PRJNA419524 (https://www.ncbi.nlm.nih.gov/bioproject/PRJNA419524/) (accessed on 1 April 2022), PRJNA493096 (https://www.ncbi.nlm.nih.gov/bioproject/PRJNA493096/) (accessed on 1 April 2022), PRJNA494633 (https://www.ncbi.nlm.nih.gov/bioproject/PRJNA494633/) (accessed on 1 April 2022), PRJNA573045 (https://www.ncbi.nlm.nih.gov/bioproject/PRJNA573045/) (accessed on 1 April 2022), PRJNA601736 (https://www.ncbi.nlm.nih.gov/bioproject/PRJNA601736/) (accessed on 1 April 2022), PRJNA623895 (https://www.ncbi.nlm.nih.gov/bioproject/PRJNA623895/) (accessed on 1 April 2022), PRJNA629087 (https://www.ncbi.nlm.nih.gov/bioproject/PRJNA629087/) (accessed on 1 April 2022), PRJNA671740 (https://www.ncbi.nlm.nih.gov/bioproject/PRJNA671740/) (accessed on 1 April 2022), PRJNA779483 (https://www.ncbi.nlm.nih.gov/bioproject/PRJNA779483/) (accessed on 1 April 2022), PRJNA189842 (https://www.ncbi.nlm.nih.gov/bioproject/PRJNA189842/) (accessed on 1 April 2022), PRJNA639353 (https://www.ncbi.nlm.nih.gov/bioproject/PRJNA639353/) (accessed on 1 April 2022). All bioprojects from which the experimental data were retrieved are presented in Appendix A, with the bioproject accession number, reference, and study information and citation [14,22,23,37,38,39,40,41,42,43,44,45,46,47,48,49,50,51,61,62,63,64,65,66,67,68,69,70,71,72,73,74,75,76,77,78,79,80,81,82,83,84,85].

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
