# Peer review of "Exploring the Complexity of the Human Respiratory Virome through an In Silico Analysis of Shotgun Metagenomic Data Retrieved from Public Repositories"

_viruses, 2024, doi:10.3390/v16060953_

Round 1

Reviewer 1 Report

Comments and Suggestions for Authors

The study aims, broadly, to better understand viral diversity found in respiratory samples that have been previously analyzed for microbiome/virome with the use of a newly developed pipeline. It is not clear if the main objective is to describe the human respiratory virome as such, or to analyze the usefulness of the newly developed pipeline to analyze sequence data. This is of relevance to assess the presented results as well as the scope and limitations of the data. It would be helpful to indicate in the Introduction which of these is the main aim of the study and, in the Discussion section mention the “success” in accomplishing this.

Specific comments.

1. Material and Methods and Results. Bioprojects included in the study. Several issues regarding the samples included in the study, which may affect the results and interpretation of the study, are not clear. The main questions for some Bioprojects are outlined below:

A) Two Bioprojects (PRJEB7454 and PRJNA189842)  included in the present study was carried out in samples from mummies. While that study is very interesting, consideration should be given to comparability of results obtained from mummified samples (reportedly 200 years old) with clinical samples obtained specifically for microbiome/virome analyses, as conservation and representativeness of RNA viruses detected in those samples could be compromised. As observed in Figure 9E, samples from this study have a different “behavior” than clinical samples, making it difficult to determine if the differences are a result of the type of sample, the type of underlying disorders (if any), differences in prevalent microorganisms at the time of death of the studied individual, among other possible factors. Another question regarding these samples is how the disease state of each individual was ascertained; a description of the individuals analyzed in Bioproject PRJEB7454 (Fletcher et al. https://discovery.ucl.ac.uk/id/eprint/628/1/618.pdf) in addition to persons showing evident signs of tuberculosis, some cases in which the cause of death may not have been related to tuberculosis (and for whom the specific disease state may be difficult to know) are described, such as “a woman of 26 years whose pleura and hair were found to be MTB-positive and who died 2 weeks after childbirth” and “a well-nourished 95-year-old woman (Fig. 6), had a small calcified lesion visible on radiograph in the left hilar region of the lung (Fig. 7). This was the only site to give a positive result for MTB DNA, as her right lung and abdomen samples were negative. It is suggested, therefore, that this individual and the 18 others who were MTB-positive at only one of multiple sites examined, had latent rather than active MTB infection”. Taking all of this into consideration, I think that inclusion of this Bioproject may have a significant impact on some of the “practical” conclusions that could be derived from the present analysis. As such, consideration should be given to exclusion of this dataset from the report. Alternatively, the above mentioned issues, among other potential limitations of analysis of these samples.

B) Several Bioprojects include repeated samples from the same person (for example, Biooprojects PRJEB32062; PRJNA316588; PRJNA369654). It is not clear how were samples from the same patient analyzed? Were they considered as a single person or as individual persons? How were differences at the moment of sampling taken into account? (for instance, some samples were obtained during CF exacerbations, while others were obtained when the patient did not present an exacerbation). In addition, if repeated samples from the same individual were used, did the authors find differences in virome between samples obtained at different times? Also, in the analysis of results of samples included in Bioproject PRJEB7454 (Fletcher et al. https://discovery.ucl.ac.uk/id/eprint/628/1/618.pdf), the authors indicate that there was a difference in the frequency of detection of Mycobacterium tuberculosis from samples obtained from the right and left lung of studied individuals; were there differences in virome results in samples obtained from different sites in the same individual?

2. Results. Lines 211-216. The reported numbers are confusing. First, the authors indicate that viral contigs were detected successfully in 201 samples; subsequently, they indicate that among samples with successful viral contigs, there were 529 patients represented. How was the number of patients obtained? From the information in the Bioprojects, most samples belong to a single person and, in many cases, several patients have repeated samples; as such, the number of patients included in the Bioprojects in less than the number of samples. Please clarify.

3. Results. In line with the previous comments, can the authors indicate how the 201 samples that successfully provided viral contigs were “divided” according to the Bioproject and how many individual patients were included in each of these Bioprojects (taking into account that in several projects some patients had repeated samples). A Supplementary table (such as Table S2) could be included with only the 16 projects that provided results and, in addition to the column in which the number of subjects included in the study is noted, additional columns with the number of samples and subjects that were included from each Bioproject indicated. For instance, in Bioproject PRJNA779483 there were 411 children studied; how many samples of these 411 children were included in the final dataset? Also, in Results section (line 217) the authors indicate that analyzed samples derived from 13 disease cohorts; taking into consideration that there were 16 Bioprojects included in the study, could the authors clarify how these 13 cohorts are conformed? I understand that in the case of Bioprojects PRJEB7454 and PRJNA189842, samples were derived from the same “cohort” (mummies from Hungary); is the reduction of 16 Bioporjects to 13 cohorts result of several Bioprojects from the same cohort or some Bioprojects ultimately not providing any successful sequences?

4. Results and Supplementary Figure S2. The number of samples included in Panel A appear to consider all samples from the selected Bioprojects and constast with panels B and c which appear to include only those with successful sequences. Please revise and include only data from samples in which successful sequences were obtained to better understand the results of the study. Also, some samples in some Bioprojects do not indicate patient’s age. Were the authors able to identify the age of all subjects represented in the study samples or were there some “unknowns”? If so, please indicate this.

5. Results and Discussion. Most of the studies identified initially were excluded and, subsequently, most selected studies and most selected samples were either excluded or did not provide adequate results. This can lead to certain biases (which would be difficult to assess) on the samples ultimately included in the study. Please comment on this.

6. Results. How did the results obtained in the current study compare to results reported by researchers on each of the analyzed studies? The authors report results of analyses carried out with the EVEREST platform. It would be of relevance to better understand the results of the study and the results obtained with this platform to compare the results of the present study with those obtained by the authors in each of the original studies: A) Can the “success” rate of viral identification using the novel platform be compared to results obtained from the original studies? B) Can the authors provide information regarding results obtained from each individual study and compare them to results obtained in the original studies?

6. Results. Figures comparing diversity between populations, types of samples and diseases. Could some differences found between disease groups be driven by the population type (for instance, sarcoidosis being conformed by adult patients, while other types of samples may represent children more frequently? Please comment on this.

7. In the Discussion. The authors note that there may be changes with age in the virome composition (lines 448-449). Taking into consideration that the largest proportion of viruses identified in the study were bacteriophages, could these changes, reflect phages that may be associated with certain bacteria which are more common at certain ages (for instance Moraxella, Corynebacterium, Staphylococci, etc.). Has this been assessed? Can the authors comment on this.

8. Results. Discussion. In line with the previous comment, to what extent, the finding in some cases, such as CF patients, are affected by the colonization of certain bacteria (such as Pseudomonas sp. or Stenotrophomonas maltophilia) and, as a consequence, by bacteriophages associated to these microorganisms? Has this been assessed? Can the authors comment on this.

9. Discussion section. The authors discuss the finding of CrAssphages in samples included in the study (lines 409-419). However, this finding is not mentioned previously in the text. The authors should include some mention of this on the Results section, so that mention in the Discussion section does not appear “unexpectedly”.

Minor comments.

1. Supplementary Figure S2. In Figure legend, the authors indicate “Demographics of the studies analysed in this chapter.”. Please revise text, as the use of the term “chapter” appears to be derived from a Thesis, since for the article the sections are not divided in Chapters.

2. Figure 8. There are some mistakes. For instance, panel B legend indicates Adults vs Upper (instead of Adult vs. Children); and Panel E is labeled as Panel F. Please revise and correct.

Author Response

REVIEWER 1.

The study aims, broadly, to better understand viral diversity found in respiratory samples that have been previously analyzed for microbiome/virome with the use of a newly developed pipeline. It is not clear if the main objective is to describe the human respiratory virome as such, or to analyze the usefulness of the newly developed pipeline to analyze sequence data. This is of relevance to assess the presented results as well as the scope and limitations of the data. It would be helpful to indicate in the Introduction which of these is the main aim of the study and, in the Discussion section mention the “success” in accomplishing this.

RE: The purpose of our study was to reanalyse previously published and publicly available shotgun metagenomics datasets obtained from respiratory samples with the objective of providing a more comprehensive understanding of the viral diversity associated with the respiratory tract. Although our methodological approach involved a pipeline recently developed by one of the authors, our objective was not to validate this pipeline, as this would require a completely different study approach. We have made this clearer in the introduction section as requested by this reviewer. Following another comment from this reviewer (see below), we are also providing a comparison between EVEREST findings and the findings reported in those specific studies in the new Supplemental Table 3, demonstrating that our processing pipeline performed well in capturing and classifying sequences that were reported in the studies inputted in EVEREST. This new information is discussed in both the result and discussion sections of the revised version of the manuscript. The “success” in accomplishing the aim of our study is discussed in the revised version of our manuscript.

Lines 117-120 (introduction):

The aim of this study was to collate publicly available sequence data from human respiratory samples and reanalyze them with the objective of providing a more comprehensive understanding of the viral diversity associated with the respiratory tract.

Lines 114-1116 (discussion):

In this study, we reanalysed sequence data files from biological specimens obtained from the human respiratory tract to better understand the viral composition and diversity associated with the human airways.

Specific comments.

  1. Material and Methods and Results. Bioprojects included in the study. Several issues regarding the samples included in the study, which may affect the results and interpretation of the study, are not clear. The main questions for some Bioprojects are outlined below:
  2. A) Two Bioprojects (PRJEB7454 and PRJNA189842) included in the present study was carried out in samples from mummies. While that study is very interesting, consideration should be given to comparability of results obtained from mummified samples (reportedly 200 years old) with clinical samples obtained specifically for microbiome/virome analyses, as conservation and representativeness of RNA viruses detected in those samples could be compromised. As observed in Figure 9E, samples from this study have a different “behavior” than clinical samples, making it difficult to determine if the differences are a result of the type of sample, the type of underlying disorders (if any), differences in prevalent microorganisms at the time of death of the studied individual, among other possible factors. Another question regarding these samples is how the disease state of each individual was ascertained; a description of the individuals analyzed in Bioproject PRJEB7454 (Fletcher et al. https://discovery.ucl.ac.uk/id/eprint/628/1/618.pdf) in addition to persons showing evident signs of tuberculosis, some cases in which the cause of death may not have been related to tuberculosis (and for whom the specific disease state may be difficult to know) are described, such as “a woman of 26 years whose pleura and hair were found to be MTB-positive and who died 2 weeks after childbirth” and “a well-nourished 95-year-old woman (Fig. 6), had a small calcified lesion visible on radiograph in the left hilar region of the lung (Fig. 7). This was the only site to give a positive result for MTB DNA, as her right lung and abdomen samples were negative. It is suggested, therefore, that this individual and the 18 others who were MTB-positive at only one of multiple sites examined, had latent rather than active MTB infection”. Taking all of this into consideration, I think that inclusion of this Bioproject may have a significant impact on some of the “practical” conclusions that could be derived from the present analysis. As such, consideration should be given to exclusion of this dataset from the report. Alternatively, the above mentioned issues, among other potential limitations of analysis of these samples. 

RE: We agree with the concerns raised by this reviewer regarding intrinsic variability associated with these specific sequencing files. Based on the data provided in both manuscripts, the FASTQ files associated with these 2 bioprojects were treated as “positive for Mycobacterium tuberculosis infection”. We did not distinguish between latent or active infections in the analyses shown in Figure 8-9 (Figures 7,8, S4 and S5 in the revised version of the manuscript) and agree with the reviewer that the virome associated with both disease states might be different and therefore confound the metanalyses. We have therefore decided to remove these samples from the diversity-related analyses shown in Figures 8-9. We have modified the result section accordingly highlighting the reason why the viral profiles from these FASTQ files were excluded from the metanalysis.

Lines 608-613 (results):

We next explored the viral community profiles obtained through EVEREST in relation to covariates such as disease state, respiratory niche, or the type of biospecimen used to sample the airways. We excluded from these analyses the profiles obtained from the two reports studying 18th-century mummified lung tissue from subjects infected with Mycobacterium tuberculosis , as representativeness of viral particles and nucleic acids in these samples may be compromised due to their age of over 200 years [38,50].

  1. B) Several Bioprojects include repeated samples from the same person (for example, Biooprojects PRJEB32062; PRJNA316588; PRJNA369654). It is not clear how were samples from the same patient analyzed? Were they considered as a single person or as individual persons? How were differences at the moment of sampling taken into account? (for instance, some samples were obtained during CF exacerbations, while others were obtained when the patient did not present an exacerbation). In addition, if repeated samples from the same individual were used, did the authors find differences in virome between samples obtained at different times? Also, in the analysis of results of samples included in Bioproject PRJEB7454 (Fletcher et al. https://discovery.ucl.ac.uk/id/eprint/628/1/618.pdf), the authors indicate that there was a difference in the frequency of detection of Mycobacterium tuberculosis from samples obtained from the right and left lung of studied individuals; were there differences in virome results in samples obtained from different sites in the same individual?

RE: We did not analyse data using a longitudinal approach because after running EVEREST, we only obtained longitudinal viral data for 3 subjects from the bioproject PRJEB32062. Additionally, sampling in these patients mostly occurred with long time intervals between each sample. EVEREST did not produce longitudinal data from FASTQ files in bioproject PRJNA316588, and paired samples in bioproject PRJNA369654 are complementary as they were used to target different viral populations (e.g. RNA and DNA) (see Tables S3 and S4 in the revised version of the manuscript). In the case of PRJEB32062, we did not consider the clinical presentation of the patients at sampling (e.g. exacerbation versus non-exacerbation). We have however considered this and other comments by the reviewer and have reanalysed the viral profiles in the metanalysis section and generated new Figures 7, 8, S4 and S5. Analysis of alpha diversity using viral profiles from bioproject PRJEB32062 is shown in the new supplemental Figure S4. Analysis of alpha diversity using viral profiles from bioproject PRJNA369654 is shown in the new supplemental Figure S5.

Regarding the question on bioproject PRJEB7454, not all the FASTQ files associated with this bioproject yielded results in EVEREST (see Table S3 and S4 in the revised version of the manuscript. Accordingly, only two FASTQ files (ERR650569 and ERR651001) from the same mummy (body 68) representing both left lung extracts were successful in EVEREST (Table S3-S4). In both FASTQ files, EVEREST identified phages only, including Gordonia and Mycobacterium associated phages.

  1. Results. Lines 211-216. The reported numbers are confusing. First, the authors indicate that viral contigs were detected successfully in 201 samples; subsequently, they indicate that among samples with successful viral contigs, there were 529 patients represented. How was the number of patients obtained? From the information in the Bioprojects, most samples belong to a single person and, in many cases, several patients have repeated samples; as such, the number of patients included in the Bioprojects in less than the number of samples. Please clarify.

RE: We thank the reviewer for bringing this to our attention, as this was the result of using the word “sample” interchangeably when referring to both “FASTQ files” and “true biological specimens”. As this reviewer pointed in the following comment, some FASTQ files represent pooled samples from different individuals, thus we agree that the use of “sample” is not appropriate. We have carefully revised the manuscript to avoid any potential confusion associated with the terminology used. We have also included a new Table S3 in supplemental material containing detailed metadata linked to the FASTQ files from which we obtained viral contigs. During the process of generating the Supplemental Table 3, we became aware of 8 FASTQ files associated with Bioproject PRJNA392272, which represent instrument technical extraction controls. In the previous version of the manuscript these FASTQ files were incorrectly grouped together with the sequencing files obtained from samples from healthy control subjects. We have excluded these samples in all the analyses presented in the revised version of this manuscript. We apologise for this mistake and have revised the manuscript accordingly including new Figures 3-6 and Figures S2-S3.

  1. Results. In line with the previous comments, can the authors indicate how the 201 samples that successfully provided viral contigs were “divided” according to the Bioproject and how many individual patients were included in each of these Bioprojects (taking into account that in several projects some patients had repeated samples). A Supplementary table (such as Table S2) could be included with only the 16 projects that provided results and, in addition to the column in which the number of subjects included in the study is noted, additional columns with the number of samples and subjects that were included from each Bioproject indicated. For instance, in Bioproject PRJNA779483 there were 411 children studied; how many samples of these 411 children were included in the final dataset? Also, in Results section (line 217) the authors indicate that analyzed samples derived from 13 disease cohorts; taking into consideration that there were 16 Bioprojects included in the study, could the authors clarify how these 13 cohorts are conformed? I understand that in the case of Bioprojects PRJEB7454 and PRJNA189842, samples were derived from the same “cohort” (mummies from Hungary); is the reduction of 16 Bioprojects to 13 cohorts result of several Bioprojects from the same cohort or some Bioprojects ultimately not providing any successful sequences?

RE: Following the suggestion by this reviewer we have included the requested information in Table 1 (general description of the bioprojects from which EVEREST successfully recovered viral contigs). A new table in supplemental material (Table S3) contains a more comprehensive description of these projects. Table S3 also contains information regarding the correspondence between the viral profile obtained through EVEREST, and the findings reported by the authors of each of the studies (following another request by this reviewer). We have also included a new table S4 in supplemental material containing the taxonomic paths and RPKM data output from EVEREST for every FASTQ file from which we obtained viral contigs.

Regarding bioproject PRJNA779483, each FASTQ file represent pooled samples (11-12 pooled throat swabs extract per FASTQ file). As shown in Table S3, EVEREST detected viral contigs in 33 out of the 37 FASTQ files representing a total of 364 throat swabs.

Regarding the discrepancy between the number of bioproject (16), and the number of cohorts (13) reported in our manuscript. This is the result of using “disease cohort” to refer to different disease groups not to the different bioprojects (e.g. some bioprojects such as PRJNA629087 and PRJNA419524 contained information from LTR subjects). We agree that the use of “disease cohort” might lead to a misunderstanding, so we have carefully revised the manuscript and use the term “clinical phenotype” when referring to the different disease states represented in each of the bioprojects that yielded results through EVEREST.

  1. Results and Supplementary Figure S2. The number of samples included in Panel A appear to consider all samples from the selected Bioprojects and contrast with panels B and c which appear to include only those with successful sequences. Please revise and include only data from samples in which successful sequences were obtained to better understand the results of the study. Also, some samples in some Bioprojects do not indicate patient’s age. Were the authors able to identify the age of all subjects represented in the study samples or were there some “unknowns”? If so, please indicate this.

RE: Panel A in Supplemental Figure 2 represents the number of individuals that were represented in the FASTQ files from which EVEREST detected viral contigs. We grouped these individuals into two broad categories: “children” and “adults”, and within these two categories we distinguished between those that were healthy and those that were described with a clinical condition. Following another question by this reviewer, we have included patient age in the new Supplemental Table 3 for those FASTQ files for which this was possible accordingly with the information provided in the original publication. As the reviewer can observe for most patients, this value was missing. We therefore decided to group subjects from these studies into two major categories: adult and children.

Panels B and C in Supplemental Figure 2 represent the number of FASTQ files from which EVEREST detected viral contigs. In panel B, we grouped these FASTQ files by clinical phenotype including healthy individuals. In panel C, we grouped these FASTQ files by the type of biological specimen and coloured the different types of biological specimens based on if they represent the upper or the lower airways.

We have revised the figure legends in Supplemental Figure 2 to make this clearer for the readers.

  1. Results and Discussion. Most of the studies identified initially were excluded and, subsequently, most selected studies and most selected samples were either excluded or did not provide adequate results. This can lead to certain biases (which would be difficult to assess) on the samples ultimately included in the study. Please comment on this.

RE: The inclusion and exclusion criteria were established in alignment with the objective of our study. These filters were implemented to select the published studies and available datasets with the highest quality and relevance in line with the objectives of our study, however some level of bias is inevitable. As the reviewer pointed, not all the FASTQ files yielded successful results, and this could be associated with the stringent filters that EVEREST implements. However, our objective was to increase our understanding of the viral diversity associated with the human respiratory tract, not to provide a definite picture of the virome associated with the human respiratory tract in health and disease. Thus, even if some FASTQ files were not successful, we are still providing the scientific community with valuable evidence, such as the previously undescribed high abundance and diversity of bacteriophages including CrAssphages in the human respiratory tract. In the case of the metanalysis, we acknowledge in the manuscript that there are intrinsic limitations to these analyses, and caution should be used in interpreting the results. We consider that the results of the metanalysis should be used as “hypothesis generating”, which should be addressed with studies specifically designed to evaluate these hypotheses.

  1. Results. How did the results obtained in the current study compare to results reported by researchers on each of the analyzed studies? The authors report results of analyses carried out with the EVEREST platform. It would be of relevance to better understand the results of the study and the results obtained with this platform to compare the results of the present study with those obtained by the authors in each of the original studies: A) Can the “success” rate of viral identification using the novel platform be compared to results obtained from the original studies? B) Can the authors provide information regarding results obtained from each individual study and compare them to results obtained in the original studies?

RE: We thank the reviewer for this comment. We have included a table in supplemental material (Table S3) in the revised version of our manuscript including information regarding the viral findings reported in the original studies, and the annotation of the viral contigs identified by EVEREST. As the reviewer can observe, we obtained a good correspondence between the viral contigs identified by EVEREST and those reported in the original studies. For all the studies, EVEREST observed a higher viral diversity than in the original studies. There was one study in which EVEREST did not identify the same viruses as those reported in the original manuscript. However, this specific study used accessed the NCBI database in 2013, while EVEREST uses the 2022 NCBI database, which might explain these discrepancies. We have included a new paragraph in the revised version of our manuscript discussing these comparisons.

Lines 1133-1135 (discussion):

In general, EVEREST reproduced the viral profiles reported in the original studies, except for one study in which EVEREST did not captured contigs classified within the Anelloviridae family. However, while the original study accessed the NCBI database in 2013 [40], EVEREST uses a more updated version [35], which might explain these discrepancies.

  1. Results. Figures comparing diversity between populations, types of samples and diseases. Could some differences found between disease groups be driven by the population type (for instance, sarcoidosis being conformed by adult patients, while other types of samples may represent children more frequently? Please comment on this.

RE: We discussed this possibility in the previous version of the manuscript (“However, it is important to exercise caution interpreting these findings, as the limited number of available studies and differences in airway sampling methodologies within the various cohorts could have potentially confounded our observations.”) and agree with the reviewer that it could be a potential explanation. However, we did not observe differences in viral diversity between FASTQ files associated with samples obtained from children and adults (see Figure 8B in the revised version of the manuscript).

In the new version of the manuscript, we have clearly discussed the limitations associated with the interpretation of the results presented in the revised version of Figures 7 and 8.

Lines 1313-1322 (discussion):

Our study has several limitations. Firstly, the small number of studies and dataset available at the time of analysis, coupled with the highly heterogenous population related to both the clinical presentation and respiratory niche sampled, represents a challenge. Thus, it is important to exercise caution when interpreting the findings of our study, as the limited number of available studies and differences in airway sampling methodologies across the various cohorts could have potentially confounded our observations. Furthermore, most of the analysed studies targeted DNA, with only one study targeting both RNA and DNA populations. Consequently, our study is biased towards DNA viruses, likely underestimating the true diversity of the virome associated with the human respiratory tract.”

  1. In the Discussion. The authors note that there may be changes with age in the virome composition (lines 448-449). Taking into consideration that the largest proportion of viruses identified in the study were bacteriophages, could these changes, reflect phages that may be associated with certain bacteria which are more common at certain ages (for instance Moraxella, Corynebacterium, Staphylococci, etc.). Has this been assessed? Can the authors comment on this.

RE: We thank the reviewer for this interesting comment. Although healthy lungs seem to harbour a transient rather than a stable microbiota like in other body sites, most of the successful sequencing files represent samples from individuals diagnosed with a respiratory condition. Respiratory conditions can affect the mucociliary clearance mechanism, or in the case of most chronic respiratory disorders, associated comorbidities such as gastrointestinal symptoms (e.g. gastrointestinal reflux) might increase the bacterial load accessing the airways. Taking all together, it is likely that the bacteriophages observed by EVEREST are associated with the bacterial population accessing the airways. We have discussed this possibility in the revised version of our manuscript. The bacterial component of the respiratory microbiota has been comprehensively characterized across the existing literature, and we did not evaluate the correspondence between phages and bacteria as it was beyond the scope of our study. However, we have included the taxonomic profiles and associated RPKM values for each of the FASTQ files that yielded results in EVEREST in a new table in supplemental material (Table S4).

  1. Results. Discussion. In line with the previous comment, to what extent, the finding in some cases, such as CF patients, are affected by the colonization of certain bacteria (such as Pseudomonas sp. or Stenotrophomonas maltophilia) and, as a consequence, by bacteriophages associated to these microorganisms? Has this been assessed? Can the authors comment on this.

RE: As indicated in our previous response and stated in the revised version of our manuscript (lines 1138-1140), the more plausible explanation for the high number and diversity of bacteriophages identified in our study is the presence of specific bacterial host. Indeed, we observed Mycobacterium phages in both bioprojects (PRJEB7454 and PRJNA189842) containing sequencing files from mummies infected with Mycobacterium tuberculosis. Likewise, EVEREST detected Pseudomonas, Staphylococcus or Ralstonia phages in bioprojects such as PRJEB32062, which contains sequencing data from people with cystic fibrosis. However, we did not explore the correspondence between specific phages and bacterial populations as this was not one of the objectives of our study. We are providing a new table in supplemental material (Table S4), which contains the taxonomic profiles associated with each of the FASTQ files that provided results with EVEREST.

  1. Discussion section. The authors discuss the finding of CrAssphages in samples included in the study (lines 409-419). However, this finding is not mentioned previously in the text. The authors should include some mention of this on the Results section, so that mention in the Discussion section does not appear “unexpectedly”.

RE: We thank the reviewer for this comment and agree that the finding of CrAssphages should have been indicated in the results section of our manuscript. We have included the following paragraph in the revised version of our manuscript.

Lines 493-497 (results):

EVEREST also annotated 27 contigs as CrAssphages or CrAss-like phage (Table S4), a type of bacteriophages commonly observed in human faecal metagenomes [53]. These contigs were associated with 6 FASTQ files from 3 different bioprojects (PRJNA392272, PRJNA419524 and PRJNA494633) (Table S4) [14,41,43].

Minor comments. 

  1. Supplementary Figure S2. In Figure legend, the authors indicate “Demographics of the studies analysed in this chapter.”. Please revise text, as the use of the term “chapter” appears to be derived from a Thesis, since for the article the sections are not divided in Chapters.

RE: We thank the reviewer for bringing this to our attention. We have changed “chapter” for “study” in the revised version of the manuscript.

  1. Figure 8. There are some mistakes. For instance, panel B legend indicates Adults vs Upper (instead of Adult vs. Children); and Panel E is labeled as Panel F. Please revise and correct.

RE: We thank the reviewer for bringing this to our attention. Figure 8 (Figure 7 in the revised version of the manuscript) has been modified following previous comments by this reviewer. We have carefully revised legends in the new Figures.

Reviewer 2 Report

Comments and Suggestions for Authors

Can authors make distinctions in the graphs between studies based on RNAseq (RNA as source of the data) and DNA so that distribution of pathogens is a bit more clear and correlated with source material? Why there are so many phages? Why authors kept studies based on DNA? What type of viral pathogens they expected to be associated with disease, and did they find them?

Add in figure 2 what are the inclusion and exclusion criteria

Can authors describe more in detail the pipeline? What programs are used for quality check,  trimming, filtering, reference-based or de novo assembly etc.. this information is provided in supplementary figure but I think it’s important to have it included in the text.

Author Response

  1. Can authors make distinctions in the graphs between studies based on RNAseq (RNA as source of the data) and DNA so that distribution of pathogens is a bit more clear and correlated with source material?

RE: The number of studies/datasets based on DNA meeting the inclusion/exclusion criteria was higher than those targeting RNA viruses, with only one bioproject (PRJNA369654) targeting both populations. As a result, our study is biased towards DNA viruses. Thus, making distinctions between DNA and RNA viral populations will be influenced by the specific clinical conditions or the respiratory niches sampled in those studies targeting RNA. Consequently, we combined the results from both viral populations and have acknowledged that our characterization of the respiratory virome is biased towards DNA viruses when discussing the limitations of our study (lines 1313-1322 in the revised version of the manuscript). We have also included the source material for each bioproject in the supplemental material in the new Table S3.

  1. Why there are so many phages?

RE: As indicated in our previous response to a similar question by Reviewer 1 and stated in the revised version of the manuscript (lines 1138-1140), the more plausible explanation for the high observed diversity of bacteriophages is the presence of specific bacterial host. However, we did not explore the correspondence between specific phages and bacterial populations, as this was not an objective of our study. We are providing a new table in supplemental material (Table S4), which contains the taxonomic profiles associated with each of the FASTQ files that provided results with EVEREST.

  1. Why authors kept studies based on DNA?

RE: We kept studies that met the inclusion/exclusion criteria targeting both DNA and RNA viruses to account for the nucleic acid diversity of viruses (see new Figure 2 in the revised version of our manuscript). However, the number of studies/databases targeting DNA viruses was higher than those targeting RNA viruses, with only one bioproject (PRJNA369654) targeting both RNA and DNA viral populations (see new Table S3 in supplemental material). The higher number of studies targeting DNA may be related to the typically low biomass associated with biospecimens from the respiratory tract, and/or technical challenges associated with working with RNA. Consequently, this is likely to bias our results toward the characterization of the DNA viral population in the human respiratory tract. We have discussed this potential bias as a limitation of our study.

  1. What type of viral pathogens they expected to be associated with disease, and did they find them?

RE: The purpose of our study was to provide a more comprehensive picture of viral populations associated with the viral respiratory tract. We did not evaluate the presence of specific viral pathogens associated with particular respiratory pathologies. We have evaluated the correspondence between the viral profiles reported in the original studies and those obtained through our computational pipeline, EVEREST. This information is presented in the new supplemental Table S3 and discussed in lines 1131-1135 of the revised version of our manuscript. Additionally, the taxonomic profiles obtained through EVEREST are included in the new supplemental Table S4.

  1. Can authors describe more in detail the pipeline? What programs are used for quality check,  trimming, filtering, reference-based or de novo assembly etc.. this information is provided in supplementary figure but I think it’s important to have it included in the text.

RE: We have included the requested information in the section 2.4 in the revised version of the manuscript.

Lines 204-215:

Data were processed using the bioinformatic pipeline EVEREST (https://github.com/agudeloromero/EVEREST). EVEREST is an end-to-end pipeline designed for virus discovery, structured into five main phases that uses FASTQ files as input. Briefly, during the pre-processing phase, files undergo quality control through trimming [26,27], followed by a filtering phase that includes host removal, replicated sequences elimination, and digital normalization [28-30]. Next, a de novo assembly is constructed using SPAdes [31] and similar contigs are clustered [32]. In the refinement phase, viral contigs are captured with VirSorter2 [33] and their quality is assessed using Check V [34]. Finally, during the viral classification phase, two databases are used, nucleotide (NCBI) and amino acid (Uniprot), to taxonomically classify the viral contigs [32,35]. Each process is executed by select and specific software tools organized within the pipeline itself, as illustrated in Figure S1.

Round 2

Reviewer 1 Report

Comments and Suggestions for Authors

Thanks for the opportunity to review the revised version of the manuscript. I do not have additional comments.